# Morpho-Histology and Morphometry of Chicken Testes and Seminiferous Tubules among Yellow-Feathered Broilers of Different Ages

**DOI:** 10.3390/vetsci9090485

**Published:** 2022-09-08

**Authors:** Jos Dorian Lawson Mfoundou, Yajun Guo, Zunqiang Yan, Xinrong Wang

**Affiliations:** 1College of Animal Science and Technology, Gansu Agricultural University, Lanzhou 730070, China; 2College of Animal Science and Technology, China Agricultural University, Beijing 100193, China

**Keywords:** yellow-feathered broiler, chicken testes, morpho-histology, morphometry, different age stages

## Abstract

**Simple Summary:**

Testes are important male reproductive organs that in chickens have been greatly investigated, from pre-hatch to after sexual maturity. The present study investigated the changes in components that occur during growth, and evaluated morphometry of the seminiferous tubules (ST), as well as gonadosomatic index (GSI) in *Gallus domesticus* at different age stages. The left and right testes were harvested from 70 chickens, then fixed in alcoholic acetate formalin (AAF) fixative solution, and hematoxylin- and eosin-stained tissues were used for microscopic observations. The results revealed that the left testis (LT) and the right testis (RT) exhibited fuzzy ST features, with apoptotic resorption of many tubules observed in both testes of 1-wk-old chicks only. ST formation was completed at 1 month, with an increase of all morphometric parameters in both testes until sexual maturity (3-mo-old): the age at which we recorded the greatest GSI. This study provides details on ST apoptotic resorption, which is a process not yet reported in existing publications, as well as ST morphometry and GSI, from a juvenile stage of growth towards sexual maturity. This can serve as reference material and also as a data update to better understand the morpho-histological changes that occur in chicken testes during growth.

**Abstract:**

Unlike in many mammals, poultry testes are found in the abdominal cavity where they develop and perform spermatogenesis at high body temperature. Scarce reports among current publications detail the growth of testes and ST morphometry among juvenile chicks. Therefore, this study aims to investigate changes in components occurring in *Gallus domesticus* testes, by assessing the GSI and morphologically and histologically evaluating the testes and ST morphometry from 1-wk- to 4-mo-old. Right and left testes were collected from 70 healthy chickens divided into seven age-related groups (*n* = 10) and then immersed into the alcoholic acetate formalin (AAF) fixative solution. Hematoxylin- and eosin-stained tissues were used for microscopic observations. The findings revealed that both testes exhibited smooth features from 1-wk-old to 1-mo-old, and thereafter showed a consistent increase in vascularization until 4-mo-old. Histologically, both testes exhibited unclear ST, with ST apoptotic resorption observed in the 1-wk-old chicks. Until 1-mo-old, ST formed and few spermatogonia differentiated into primary spermatocytes, with all spermatogenic cells observed at 3-mo-old, i.e., sexual maturity. These findings suggest that both testes develop in analogy, and their sizes including increases in length and diameter are related to the spermatogenic activity in the ST. Subsequently, ST resorption by apoptosis is assumed to participate in the physiological mechanism regulating germ cells (GC). Finally, the GSI tended to increase with growth.

## 1. Introduction

Testes are the male reproductive organs that have two primary functions: testosterone synthesis and sperm production. These functions are critical not just for the conservation of male traits, but also for the conservation of species [1]. Primordial germ cells (PGCs) are the earliest germ-cell population to form, and they are the shared ancestor of oocytes and spermatogonia. In chickens, the embryonic urogenital system arises from the intermediate mesoderm at about day three of fetal development and can be recognized by thickening of the coelomic epithelium ventral to the mesonephros. PGCs travel through the bloodstream from the germinal crescent to the gonads, where they increase in number with the gonads’ enlargement [2,3]. Around day 5.0–5.5 of incubation, the undifferentiated gonads yield the primary sex cords, which grow following the rete cords [4]. Those primary sex cords, which include the gonadal PGCs, spread into the gonad’s mesenchymal section, leading the gonads to grow symmetrically in males but asymmetrically in females, with only the left gonad forming at day six of incubation [2,5]. Sertoli cells (SC) assume an essential physiological function in the male reproductive system, particularly in the testes, where they sustain, nurture, and preserve GC [6]. Female GC undergo meiosis and cease at prophase I, while the male GC enter mitotic arrest and begin meiosis just after birth. Spermatogonia initiates meiosis, leading to the development of primary spermatocytes followed by secondary spermatocytes, which further grow into spermatids, while the spermatogonial stem cells (SSCs) remain unchanged; these are the cells that give rise to spermatogenesis [2,7,8].

A great range of scientific reports on testes is available; Kumaran and Turner studied the normal development of testes among White Plymouth Rock (WPR) chickens pre-hatch to sexual maturity, both morphologically and histologically [9]. King and McLelland anatomically and histologically studied the development of testes, epididymis, and vas deferens with a focus on adult chickens [5]. Razi et al. described the histology, anatomy, and morphometry of the male genital system in mature Iranian Native White Roosters (IWR) [10]. Testis morphology, histology, and morphometry have also been described in other avian species including Japanese quail [11,12], Duck [13,14], turkey [15], ostrich [8], and dove [16]. Spermatogenesis has been thoroughly investigated in mammals, and most of our understanding of the developmental and transformative mechanisms in the spermatid originates from these studies (spermatogenesis in mammals [17], birds [18,19], also in nonmammalian vertebrates [20]). GSI, a parameter used to evaluate sperm production efficiency, has been greatly used and reported for adult chickens [21,22], guineafowl (GF) [23], turkey [24], and fish [25,26], and also for certain mammals including humans [1]. Chicken testes have even been used to investigate obesity and diabetes mellitus in humans [27].

Although a large range of literature on chicken testes is currently available, studies are scarce detailing testicular morphology and histology, morphometry of the ST, or GSI in juvenile chickens towards sexual maturity. Most of the available literature has considered the subject in adults that have already reached sexual maturity and have been sexually active for a certain period.

Therefore, in this study, we aimed to investigate the morpho-histological attributes and morphometry of both testes and ST at various stages of growth, evaluate GSI, and determine the ages at which each various spermatogenic cell developed and became noticeable within the ST epithelium, among chickens aged between 1-wk-old and 4-mo-old.

## 2. Materials and Methods

### 2.1. Ethics Statement

The Animal Ethical and Welfare Committee (AEWC) of Gansu Agricultural University evaluated and approved all experimental methods included in this work (approval number: 2019-044).

### 2.2. Animals

All experimental animals used in this study were obtained from Gansu Haikang Agricultural Technology Development Co., Ltd., a broiler poultry farm in Gansu province, Lanzhou city, China. The yellow-feathered broilers (Huang Yu Rouji) are Chinese native chickens that have been bred by crossbreeding local high-quality breeds for meat production. They are well-known in Asia in general and in China in particular, especially for their unique meat. A total of 70 healthy male chickens were used in this study, all organized in groups of ten (*n* = 10) according to their developmental stages, which included chickens aged 1 week, 3 weeks, 1 month, 1.5 months (6 weeks), 2 months, 3 months, and 4 months, respectively.

### 2.3. Sample Preparation for Anatomical Observations

The preliminarily experimental procedure consisted of weighing the chickens to record their live weight, and evaluating the GSI by using Orlu and Egbunike’s formula (both testes’ weight divided by body weight and multiplied by 100) [28]. This was followed by their slaughter and the opening of their abdominal and thoracic cavities to observe and record the location and evolution of testes and their relation with the surrounding organs. Then, left and right testes were carefully collected with a record of their weight and measurements (length, width, and thickness).

### 2.4. Sample Preparation for Microscopic Observations

The testes samples were placed into the alcoholic acetate formalin (AAF) fixative solution immediately after collection and the recording of all required parameters. The fixed samples were then conditioned in wax paraffin and finally stained with hematoxylin and eosin (HE) for microscopic observations.

The LT and RT micrographs of 1-week-old chicks at various magnifications (40× to 400×) were used to evaluate ST regression by apoptosis and identify the testes region where this was found. According to the ST epithelium regression, apoptosis was classified into two types: (1) fully regressed for tubules with empty lacunae, and (2) regressing for those with a part of their epithelium that harbored a single layer of cells (spermatogonia and SC) but appeared condensed and shrank with cell irregularities, detached from the tubules myoid cells, and appeared different to the normally forming tubules.

### 2.5. Data Collection

The testes’ ST diameters, tubules’ epithelium thickness, and lumen diameters were collected using ImageJ 1.53a for Windows (ImageJ, NIH, Bethesda, MD, USA) [29]. For that purpose, histological micrographs of 400× magnification were randomly selected and used to collect the data for 30 rounded to oval tubules in all age groups studied. The method in the present study was used previously for ST morphometric data collection from domestic cats, donkeys, and mules [30,31].

### 2.6. Statistical Analysis

The statistical analyses of the data were carried out using SPSS 25.0 for Windows (SPSS Inc., Chicago, IL, USA). To ensure that the data met the requirements of the variance analysis, descriptive statistics were used to calculate the mean and standard deviations for each age group. One-way ANOVA testing was performed to examine the data, and Duncan’s multiple comparison test was employed to evaluate the differences between age groups, while the independent T test was performed to evaluate the differences in morphometric data and ST components between the LT and RT of the same age group. The data were presented as mean ± standard deviation and were regarded as significant when *p* < 0.05.

## 3. Results

### 3.1. Anatomical Observations of Chicken Testes at Different Stages of Growth

#### 3.1.1. One-Week-Old Chick

Through the abdominal dissection of the male chick, it was found that the LT and RT were high in the abdominal cavity in front of the kidney, where they were attached to the coelomic wall, the epididymis facing the dorsal aorta from both sides. The differences in their sizes and lengths could be noticed at first sight; the LT was longer than the RT, the latter appearing to have a broader anterior end that was embedded in the interior part of the liver and displaying only about two-thirds of its total length (Figure 1a). The two testes appeared smooth, yellow-creamy and slightly whiten at their anterior and posterior ends, and elongated in their shapes (Figure 2). Both were separated by the mesenteric membrane that refrained them from having direct contact and had a close relationship with the proventriculus, the junction of the caeca, and the small and large intestine which ventrally covered them. In general, the testes’ relationship with the surrounding organs and their separation by the mesenteric membrane remained unchanged and was observed in every age group studied.

Figure 2 shows the evolution of the weight, length, width, and thickness in the left and right testes in the same age group (Independent Samples *t* test), and at various stages of growth for left and right testes, respectively (one-way ANOVA).

Both testes underwent a very slow growth phase from hatch to 1-wk-old. They entered a slow growth phase at 3-wk-old, displaying the first increase of parameters that almost doubled and continued to grow until 1-mo-old, the age at which both the LT and RT showed an analogous and remarkable variation of their length (Figure 2B). An accelerated growth phase was noted between 1.5-mo-old and 2-mo-old with the second increase of various parameters. From 3-mo-old, the testes entered a fast-growth phase with the third increase of parameters, which further slightly increased towards 4-mo-old (*p* = 0.993 to *p* = 1.00).

Independent samples T testing revealed that the comparison of all parameters between the left and right testes in the same age group appeared different only for the testes’ length at 1-wk-old (** *p* = 0.003) (Figure 2B) and thickness at 1-mo-old (** *p* = 0.009) (Figure 2D); all other parameters showed no difference (*p* = 0.064 to *p*= 0.942).

#### 3.1.2. Three-Week-Old Chick

The two testes still displayed a smooth feature, yellow-creamy similar to that of the previous age. The RT displayed an anterior end that appeared larger than that of the LT and became more embedded into the interior part of the liver (Figure 1b). Both testes’ weight and morphometric data are presented in Figure 2. 

#### 3.1.3. One-Month-Old Chick

Both testes displayed slow growth relative to the preceding age points (Figure 2), the LT became slightly elongated while the anterior end of the RT seemed to have become slightly enlarged. The testes’ color remained yellow-creamy and still displayed a smooth feature on which blood vessels (BV) could now be seen growing from the epididymis to their arch section (Figure 1c). The embedding of the RT anterior end within the interior part of the liver grew further.

#### 3.1.4. One-Month and Half-Old (6 Weeks) Chick

The 1.5-mo-old testes displayed significant differences in their parameters (Figure 2). The BV growth from the epididymis continued, displaying structures that appeared slightly longer, while their color was unchanged and smooth features were apparent (Figure 1d). The anterior end of the RT appeared more embedded in the interior part of the liver as the chick grew in age. 

#### 3.1.5. Two-Month-Old Chick

The LT and RT exhibited great changes, reflected in their recorded parameters (Figure 2). Their colors became uniformly creamier. BV continued to grow, spreading branches from the epididymis to the top arch section of the testes, which still appeared smooth at the portions uncovered by the growing BV (Figure 3a,b). The RT appeared more embedded within the interior part of the liver and its anterior end further enlarged. At this age, the LT enlargement brought it into connection with the caudal end of the related lung (Figure 3a).

#### 3.1.6. Three-Month-Old Chick

The disparity between the LT and RT became easily noticeable (Figure 2). The LT lost its elongated look and appeared more oval while the RT anterior end grew significantly, displaying a smaller posterior end (Figure 3c). The two testes’ color appeared slightly whitened relative to the preceding age points, their surfaces were smooth at the portions uncovered by the BV, which grew diffusely over the whole testes and appeared more prominent on the LT than on the RT. 

#### 3.1.7. Four-Month-Old Chick

The 4-mo-old LT and RT displayed great similarities when looking at their parameters (Figure 2). Both had whitened further and occupied a greater space in the abdominal cavity due to their sizes which increased their proximity to the lungs; the embedding of the RT anterior end into the interior part of the liver was markedly noted. At this age, the LT appeared bean-shaped with anterior and posterior ends that looked almost alike, whereas the RT displayed a much broader anterior end. The vascularization had produced many branches, which spread over the whole testes for some of the collected samples, with smooth surface displayed at the uncovered portions (Figure 3d,e). 

#### 3.1.8. Gonadosomatic Index (GSI) among Chickens at Different Stages of Growth

The GSI of the growing chicken appeared stable among the chicks aged between 1-wk-old and 3-wk-old (0.02%), while the smallest value was recorded among the 1-mo-old chicks (0.01%). The time at which the testes were in the slow growth phase (Table 1) corresponded to the time the testes started to become more vascularized (1-mo-old). We recorded the greatest value for the 3-mo-old chicks (0.37%), the age at which the testes enlarged considerably (Figure 2) and showed greatest differences relative to all previous ages studied considering their sizes and the growing vascularization on their outer tunic. Thereafter a slightly decreasing trend was observed at the end of the study (0.31% at 4-mo-old).

Table 1 displays the GSI for the growing chicks at different developmental stages. The GSI appeared to be constant when the testes were still in a slow-growth phase from 1-wk-old to 3-wk-old (0.02%) and showed a decrease in its value (0.01%) at 1-mo-old, the time at which spermatocytes were firstly noticed within the tubules’ epithelium. Thereafter, it started to show an increasing trend from 1.5-mo-old to 3-mo-old (0.37%) which corresponded to sexual maturity. All spermatogenic cells became noticeable within the epithelium of the tubules. A downward trend was noted at 4-mo-old (0.31%).

### 3.2. Histological Observation of Chicken Testes at Different Stages of Growth

#### 3.2.1. Testicular Seminiferous Tubules’ Resorption by Apoptosis

ST resorption by apoptosis was observed in the 1-wk-old chick and appeared to happen simultaneously across both testes’ parenchyma, being more noted around the epididymis (Figure 4a,b). At this stage of growth, normally growing ST were forming and displayed unclear features of tubules sparsely spread across the testicular parenchyma, while some appeared to have stopped growing, therefore regressed fully and left empty lacunae on the stroma (Figure 4c,d). A second type of ST were still regressing and displayed the same components (SC and spermatogonia as a single layer of cells) as the normally emerging ST while exhibiting degenerating cells that were still regressing within the apoptotic tubules (Figure 4e,f). This appeared to happen in the 1-wk-old chick testes only and was not observed in the other ages studied thereafter until the end of the study.

#### 3.2.2. One-Week-Old Chick

The microscopic investigations of the 1-wk-old chick displayed, from the outside inward, a thin tunica albuginea rich in collagen that externally delimited the testes, the parenchyma, and the forming ST. The parenchyma of the LT displayed a sparse arrangement of the ST around the tunica albuginea, often exhibiting rounded or nearly rounded tubules that appeared to be in a more advanced stage of formation (Figure 5a) than most of those around the epididymis, which displayed elongated features (Figure 5e). The parenchyma of the RT appeared to have a divergent growth tendency, showing dense organization of the ST around the tunica albuginea and sparsely in the middle of the testis, with more rounded to oval features displayed (Figure 5b,f). Figure 6 displays the ST morphometric parameters. ST were still emerging, resulting in hazy forms and sizes, with some displaying a lumen. Basement membranes (BM) progressively formed, while the interstitial space containing isolated Leydig cells (LC) appeared broad. Spermatogonia, oval for most part, presented a single layer of cells orderly arranged near the inner part of the BM, among the SC that displayed a clear cytoplasm and a dark nucleus (Figure 7a,c).

Figure 6 shows the seminiferous tubules’ morphometric parameters during growth, in comparison between the left and right testes data in the same age group (independent T test results), and between left testes and right testes respectively (one-way ANOVA).

The tubules’ diameter and the epithelium thickness appeared to have the same growth tendency, which showed a gradual evolution of both parameters from 1-wk-old to 2-mo-old, then a great increase in their values at 3-mo-old, and which grew further until 4-mo-old. The lumen diameters appeared to have the same growth tendency as the testes (Figure 2), showing a very slow growth phase with parameters stable between 1-wk-old and 3-wk-old, followed by a slow growth phase towards 1-mo-old. An accelerated phase was observed from 1.5-mo-old toward 2-mo-old, then a fast growth phase was noted from when the chicks were 3-mo-old to the end of the study.

Independent samples T testing revealed that the left and right tubules’ diameters were different among the 1-wk-old (** *p* = 0.007) and the 3-wk-old (* *p* = 0.018) specimens, whereas the other age groups showed no difference (*p* = 0.81 to *p* = 0.412) (Figure 6A). The tubules’ epithelium in the left and right testes appeared to show a slight difference only among the 1-wk-old chicks (* *p* = 0.049), with no difference noted in the other age groups (*p* = 0.428 to *p* = 0.908) (Figure 6B); finally, the lumen diameters were only different among the 2-mo-old chicks (* *p* = 0.029), not for the other age groups (*p* = 0.057 to *p* = 0.946) (Figure 6C).

#### 3.2.3. Three-Week-Old Chick

The ST of the 3-wk-old chick testes presented clearer structures, with a denser arrangement of the tubules around the tunica albuginea than in their middle section (Figure 5c,d). Although ST could be seen developing on the parenchyma, these still exhibited unclear features with the incomplete formation of the BM by the peritubular myoid cells (Figure 5g,h), and appeared to have slightly enlarged (Figure 6), displaying various shapes and sizes. The ST exhibited a single layer of spermatogonia and SC arranged, like at the previous age point. The interstitial space was slightly reduced and displayed few isolated LC and BV (Figure 7b,d).

#### 3.2.4. One-Month-Old Chick

The 1-mo-old chick testes displayed well-formed ST features that had completed their formation, with clear delimitations by the BM of peritubular myoid cells and connective tissue. Their arrangement on both testes’ parenchyma appeared denser than precedingly, but continued to exhibit a sparse presence in their middle section (Figure 5i,j). The slight increase in the tubules’ diameter (Figure 6) consequently reduced the interstitial space that displayed larger BV and LC arranged in a small mass (Figure 5m,n). The completion of ST formation allowed meiotic entry of a few spermatogonia that differentiated into primary spermatocytes, which stained deeper than other cells and slightly moved towards the lumen for some but appeared scarcely noticeable among the ST (Figure 7e,g).

#### 3.2.5. One-Month Half-Old Chicks (6 Weeks)

At this age, the 1.5-mo-old testes displayed clearer ST (Figure 5o,p), with a denser arrangement of the tubules, and the sparse appearance medially observed had almost disappeared (Figure 5k,l); the tubules’ diameter increased slightly (Figure 6), which moved the ST slightly closer, exhibiting narrowing interstitial space containing many BV and LC arranged like those of the 1-mo-old specimens (Figure 5o,p). Few tubules exhibited more layers of spermatogonia around SC, while many displayed only a single layer as observed precedingly. Meanwhile, primary spermatocytes started becoming more noticeable (Figure 7f,h).

#### 3.2.6. Two-Month-Old Chick

At this stage of growth, the two testes displayed a denser arrangement of the ST due to their enlargement (Figure 7 and Figure 8a,b). The testes became more vascularized and exhibited larger BV, to irrigate the testes with enough blood necessary to maintain and support spermatogenesis (Figure 8a,d). Avian red blood cells (RBC) were prominent at this stage of growth displaying tiny to very large features within the intertubular compartments beside the LC (Figure 8c). The LT spermatogonia arrangement did not differ greatly relative to that at 1.5-mo-old, and exhibited a single layer of cells for many ST, while others were diffused within it (Figure 9a). Those of the RT were disseminated within the tubule’s epithelium around the SC for many ST, with primary spermatocytes noticed in greater numbers than at the previous age points (Figure 9d).

#### 3.2.7. Three-Month-Old Chick

Both 3-mo-old testes displayed a much denser arrangement of the ST with small intertubular compartments (Figure 8e,f). Meanwhile, the tubules enlarged significantly (Figure 6). It appeared that the primary spermatocytes noticed in the 2-mo-old chicks had further developed and differentiated into secondary spermatocytes, spermatids, and then spermatozoa, since all spermatogenic cells were now observed within the ST epithelium, with colonies of spermatozoa noted around the larger lumen, and individuals found within it (Figure 9b,e). The epithelium was filled with more layers of cells and no specimens displayed only a single layer as precedingly observed (Figure 8g,h). Both testes’ vascularization grew further and displayed many BV as well as the avian RBC that became prominently noticeable. The sparse look found medially in the ST had completely disappeared.

#### 3.2.8. Four-Month-Old Chick

The 4-mo-old chick testes exhibited a denser arrangement of the ST, which showed more spermatogenic activity (Figure 8i,k). The tubules were slightly enlarged relative to the 3-mo-old chicks (Figure 6) and displayed several BV, as well as prominent avian RBC within the intertubular compartments where LC were found condensed (Figure 8j,l). The ST were fully active and displayed all spermatogenic cells from the SC, spermatogonia to spermatozoa, the latter found prominently arranged similarly to those of the 3-mo-old chicks. Except for SC and spermatogonia, which remained aligned with the BM, the spermatocytes, spermatids, and spermatozoa were found migrating towards the lumen, and their differentiation was observed (Figure 9c,f).

Appendix A and all Figures in the present work can be found as Appendix A, which are openly available online.

## 4. Discussion

### 4.1. Yellow-Feathered Broilers

The yellow-feathered broiler is a native Asian chicken breed renowned for its distinctive meat taste. In China’s massive chicken consumer market, there is a growing need for flavor, and about 50% of Chinese chicken meat comes from domestically raised yellow-feathered broilers, which have a lower weight, longer production cycle, and more flavorful meat than white-feathered broilers [32,33], and in 2015 surpassed white-feathered broilers in terms of live production, reaching 4.4 billion heads [34]. These chickens are often divided into three categories based on their development rate; (1) fast growth: 60 days, (2) medium growth: 60–100 days, and (3) Slow growth: 100 days or more; the slaughtering weight is generally about 1.75 Kg [33]. The chickens used in this study were of the slow growth type, generally slaughtered at 3-mo-old.

Yellow-feathered broilers are mainly produced in the southern part of China [35]. Because South China lies in the tropical and subtropical zones, heat stress has emerged as a major issue for yellow-feathered broiler production [36], and has been shown to have a negative influence on growth performance, including decreased feed intake, poor feed efficiency, and delayed growth rates [37,38].

More research is needed on these chickens because there is little information about them publicly available online, even in the Chinese language, despite China being their primary producer.

### 4.2. Anatomo-Morphological Observation of the Testes and Seminiferous Tubule Morphometry at Different Stages of Growth

The location of both testes in the male chicks showed similarities to that of the left ovary in its female homolog: attached to the coelomic wall on either side of the dorsal aorta overlapping the two kidneys, as observed in our previous study. This falls into the same range relative to previously reported findings in other avian species, including turkey and GF [23,24,39]. Spermatogenesis, the same as oogenesis, consists mainly of three different phases; first is a period of multiplication during which the cells are known as spermatogonia; second is a period of growth during which the primary spermatocytes enlarge; and third is a period of maturation during which the first maturation division produces secondary spermatocytes, and the second maturation division produces spermatids. Each spermatid becomes a spermatozoon [5].

In this present study, we did not count the various spermatogenic cells (see [10,40]), instead focusing our attention on the changes occurring in the testes during growth. We identified the time at which each spermatogenic cell type became noticeable within the ST, and meanwhile evaluated the chicks’ GSI and ST morphometric parameters at various ages.

#### 4.2.1. One-Week-Old Chick

The testes of the 1-wk-old chicks exhibited smooth and elongated features, unlike the observation made in the Moscovy drake duck (MDD) where these appear small and oval at 1-day-old and have been estimated about 2.67 mm (~0.27 cm) L, 1 mm (~0.1 cm) Ae and Pe [14]. The mass of the testis was made up of thousands of twisted ST. Individual tubules were similar to those found in mammals, but anastomoses are significantly more prevalent in birds [5]. This study agrees with King and McLelland, and Kumaran and Turner, who stated that the surface of the testes is covered by a very thin tunica albuginea [5,9], which is what we observed throughout the whole study. In their histological study, Rizzi and Verdiglione classified the testes’ growth into four classes according to their developmental stages The first of these, Class 0, corresponds to our findings in the 1-wk-old chick: “prepubertal testis, a simple single layer of cells (spermatogonia and SC) in the ST, abundant interstitial space” [41]. Spermatocytogenesis involves mitotic divisions including spermatogonia replication and preservation [42]. The sustentacular cell (SC) stretches the entire depth of the epithelium and so supplies mechanical support for the GC while preserving tight structural and signaling interaction throughout their maturation, in order to complete spermatogenesis. It is also believed to participate in steroid hormone synthesis, meanwhile having a phagocytic ability [5,43]. At this stage, although ST remained in the formative phase which corresponds with previous reports [9], they could be seen developing on the parenchyma, whereas Lan et al. reported that few tubiform structures resembling ST were identified on 1-day-old ostrich chicks’ parenchyma, surrounded by numerous PGCs and a few spermatogonia, but with no complete ST present, similar to observations in 1-day-old White Leghorn chickens [8,44]. 

Both testes’ STs were still forming, and displayed a sparse arrangement of the ST and broad interstitial space.

#### 4.2.2. Three-Week-Old Chick

The testes at 3-wk-old exhibited no significant changes in terms of their shapes, relative to the preceding age point. The present study’s data (Figure 2) were compared with that of the 15-day-old (~2-wk-old) MDD: 3.43 mm (~0.34 cm) L, 2.00 mm (0.2 cm) Ae, and 2.17 mm (~0.22 cm) Pe for the LT, while the RT estimates were 3.33 mm (~0.33 cm) L, 2.00 mm Ae and Pe; the MDD’s data appears remarkably smaller albeit they are from different species with a week of age difference between the two [14]. The ST displayed clearer features relative to the 1-wk-old, with the same spermatogenic cell arrangement as that of Class 0 [41]. Also similar to our study, the wall of the ST was made up of a basement membrane as well as connective tissue cells and fibers, despite the age of the chicks [45], while LC were present in the gaps between the tubules. Mature LC are polygonal and have a smooth endoplasmic reticulum, tubular cristae in mitochondria, and cholesterol-rich lipid droplets which are hypothesized to be a precursor material that participates in the synthesis of androgenic hormones [5]. When type-A spermatogonia proliferate by mitotic division to develop from type-A1 to A4 cells, some type-A4 cells differentiate into intermediate spermatogonia, which divide once to produce type-B spermatogonia [43]. 

#### 4.2.3. One-Month-Old Chick

The testes of the 1-mo-old chick exhibit a smooth feature, yellow-creamy displaying the growth of vascularization from the epididymis. The recorded morphometric data at this age (Figure 2) seem to be greater than 0.5 cm L reported in the Gaga chicken (GGC) [21], while 7.57 mm (~0.76 cm) L, 4.03 mm (~0.40 cm) Ae, 2.40 mm (~0.24 cm) Pe wide for the LT and 7.43 mm (~0.74 cm) L, 3.97 mm (~0.40 cm) Ae wide, and 2.47 mm (~0.25 cm) Pe wide for the RT among the MDD [14]. The immature or inactive testes of most domestic chicken breeds and many seasonal birds are yellow due to lipid buildup in the interstitial cells. The testicular artery, a branch of the cranial renal artery, carries the majority of arterial blood to the testis in domestic fowls and is drained by several small testicular veins into the caudal vena cava [5]. At this stage of growth, few spermatogonia differentiate into primary spermatocytes and migrate towards the lumen, which corresponds to Class 1 in Rizzi and Verdiglione’s classification [41], the only difference being that we still only observed a single layer of cells within the ST than 2 as in their report, and agrees with Steger’s findings [42]. Tetraploid primary spermatocytes are formed from type B cells through one mitotic division and then undergo meiosis I to form diploid secondary spermatocytes [43]. Chicken testes seem to develop faster than those of the ostrich in which PGCs differentiation into spermatogonia was reported at this age [8]. Our findings agree with those of Lan et al., stating that the integrity of ST was found in the testis of 30-day-old (1-mo-old) ostrich chicks [8]; however, their forms were uneven, which is controversial to our observation showing well-formed ST. 

#### 4.2.4. One-and-a-Half-Month- (6 Weeks-) Old Chicks 

Morphologically, the testes of the 1.5-mo- (6-wks-) old chicks showed no great differences relative to those of the 1-mo-old except when looking at their parameters (Figure 2), and their vascularization which had grown slightly further. At this age, the LT appeared to be larger than the RT. Histologically, the ST formation continued and many tubules displayed more layers of cells, corresponding to Class 1 in the description made by Rizzi and Verdiglione: “[two] layers of cells in the ST, primary spermatocytes (cells with darkly stained chromatin) moving away from the basement membrane” for some tubules, and for others Class 2: “[three to four] layers of cells in the ST” [41]. Typical ST appeared earlier in the chicken testes than those of ostriches, at 45-days-old (~1.5-mo-old), at which point the tubal wall comprised a basement layer made up of myoid cells and collagen fibers, and a seminiferous epithelium that was made up of SC and one or two layers of spermatogenic cells [8]. 

In chickens, 1-mo-old seems to be a crucial age period during which some important events take place in the developing gonads. Spermatogonium differentiates and starts displaying primary spermatocytes in the male chicks, while in female chicks, oogonium differentiates into primary oocyte as observed in our previous study, as well as in other reported manuscripts [39,46,47].

It appears that between 1-wk-old and 1-mo-old, the testes’ development is mostly consists of ST formation, starting to display spermatocytes when the BM of ST is completely formed.

#### 4.2.5. Two-Month-Old Chick

The testes of the 2-mo-old chick displayed a creamy yellow uniform surface on which BV continued to grow (Figure 2), and appeared to be larger than those of the GGC: 0.78 cm L [21], and MDD: 11.03 mm (~0.11 cm) L, 4.70 mm (~0.47 cm) Ae wide, and 3.73 mm (~0.37 cm) Pe wide for the LT, with RT estimates of 4.67 mm (~0.47 cm) L in the same age frame [14]. At this age, many ST of the LT still exhibited a single layer of cells and some were spread around; in contrast, the RT displayed more layers of cells within the ST with many spermatocytes noted in almost all the ST, corresponding to Class 3 as classified by Rizzi and Verdiglione: “[five to six] layers of cells in the ST, presence of sperm cells demonstrating the spermatogenic activity” [41]. 

#### 4.2.6. Three-Month-Old Chick

The 3-mo-old chick testes exhibited a bean shape (LT) and a wide Ae (RT), with a smooth appearance at the sections uncovered by the spreading vascularization, and were gradually whitening. Both testes’ morphometric parameters (Figure 2) were greater than those of the GGC 1.23 cm L [21] and the MDD: 16.40 mm (~1.64 cm) L, 6.30 mm (~0.63 cm) Ae wide, 6.37 mm (~0.64 cm) Pe wide for the LT, while the RT estimates were 14.33 mm (~1.43 cm) L, 8.10 mm (~0.81 cm) Ae wide, and 5.77 mm (~0.58 cm) Pe wide [14]. With sexual activity, the testes expand and become whiter due to the scattering of LC by the enlarging ST [5]. The testes at this stage of growth could be classified into Class 4: “more than [seven] layers of cells in the ST, presence of numerous sperm cells demonstrating the active spermatogenesis in the lumen, reduction of interstitial tissue” [41]. Our findings follow those of Gerzilov et al. in 3-mo-old MDD, reporting the presence of spermatozoa within the ST [14]. Spermatids undergo spermiogenesis, which comprises nucleus condensation and elongation, synthesis of the acrosome, and flagellum production, with the arising mature spermatozoa subsequently discharged into the lumen (spermiation) [43]. Hogue and Schnetzler investigated the reproductive development of Barred Plymouth Rock cockerels, stating that males attained sexual maturity as early as 16 weeks of age (4-mo-old). In pen matings, however, an adequate level of fertility was not seen until 26 weeks (6.5-mo-old) [48], whereas Parker et al. reported observing fully developed spermatids in the tubules of White Leghorn and New Hampshire cockerels at 12 weeks of age (3-mo-old) [44]. McCartney further stated that some young males could produce viable sperm as young as 11 weeks of age (~3-mo-old), and early fertility of 85% to 100% can be predicted at that period [49]. In this present study on yellow-feathered broiler chickens, sexual maturity was observed at 3-mo-old, the age at which spermatozoa became noticeable within the tubular epithelium and also the lumen of the ST. Spermatozoa grew in clusters, with their heads linked to sustentacular cells (SC) and their tails extending into the lumen [5]. However, in 90-day-old ostrich chicks (3-mo-old), Lan et al. reported only an increase in the number of primary spermatocytes located at the surface of the ST [8]. 

#### 4.2.7. Four-Month-Old Chick

At 4-mo-old, both testes appeared similar to those of the 3-mo-old, with slight growth of their vascularization and further whitening. The LT and RT in this present study (Figure 2) showed similarities to the reports by Razi et al. for adult IWRs LT: 4.43 cm L, 10.94 cm wide, and 2.15 cm thick and RT 4.9 cm L, 2.10 cm wide, and 2.20 cm thick; except for the RT width that appeared to be far greater than that we observed in this study [10]. It was also greater than those of the GGC: 1.63 cm L [21] and the MDD: 19.93 mm (~1.99 cm) L, 9.43 mm (~0.94 cm) Ae wide, and 8.10 mm (~0.81 cm) Pe wide for the LT, and 18.63 mm (~1.83 cm) L, 8.87 mm (~0.89 cm) Ae wide, and 7.87 mm (~0.79 cm) Pe wide for RT [14]; but smaller than of the Iraqi indigenous duck: 52.67 mm (~5.27 cm) L and 58.875 mm (~5.89 cm) L for the LT and RT, respectively [50]. These also appeared greater than those of the turkey: 3.72 cm, 4.40 cm, and 4.48 cm L in the three studied groups [51], but smaller when compared to that of the adult Nigerian local breed of chicken (NLBC): 5.3 cm L, 2.7 cm wide [22]. The testis length varied from about 2 cm immediately before sexual maturity to about 5.5 cm during sexual activity [5]. At this stage of growth, the testes are fully active and the ST is full of all kinds of spermatogenic cells, corresponding to Class 4 in Rizzi and Verdiglione’s testes classification, the same as the 3-mo-old chicks [41]. 

The testes displayed a great increase in size upon completion of the ST formation, coinciding with the time that spermatogenic activity accelerated and more layers of cells were present in the tubules (2-mo-old), and also at sexual maturity (3-mo-old), which continued towards 4-mo-old. Previous authors related the testes’ increase in size to the augmentation of ST length and diameters, the greater numbers of LC, and the augmentation of the sperm production capacity [5], which in our study showed a gradual increase of ST with age.

Through observation and analysis of the current results, we can assume that spermatogenic activity also plays a role in the increase of testes size, and the more active they are, the bigger they become. This could also explain the reason why seasonal birds’ testes tend to reduce significantly in size during the non-breeding season and regain their normal sizes for the breeding season, increasing up to 300 to 500 times among certain passeriforms at the nuptial phase, with resumption of spermatogenic activity which seems to be of a much shorter duration than in mammals [5].

### 4.3. Seminiferous Tubules’ Degeneration by Apoptosis during Early Seminiferous Tubule Formation

Apoptosis as a concept was first introduced by Kerr, Wyllie, and Currie in 1972 to define a physically different form of cell death with characteristics such as DNA sequencing, chromatin condensation, and membrane blebbing [52]. Cell apoptosis, or cellular self-destruction, also includes ongoing intracellular synthesis processes regulated by cellular genes [53,54]. In this study, we observed ST regression by apoptosis in the 1-wk-old chick synchronously with the ST formation. This kind of apoptosis appears to be different from other reviewed papers, reporting GC apoptosis taking place within the ST (for more reading on male mammals’ GC apoptosis, see [55,56,57,58]), and was observed throughout growth as in the ostrich chick during growth from 1-day-old to 90-day-old (3-mo-old), and which appeared to occur mostly among LC and spermatocytes similar to reports in adult mice [8,59]. During testicular development, the stem-cell factor plays a critical role in the migration, adhesion, proliferation, and survival of PGCs and spermatogonia [60]. In this present study, it appears that some among the forming ST failed to proceed with their normal growth, consequently showing signs of degeneration and starting to regress within the stroma; various shapes were displayed that were similar to the normally forming ST. This happened simultaneously in both the LT and the RT. The ST that fully degenerated left a kind of empty lacunae with shapes that varied from rounded, elongated, to curved relative to the resorbed ST; those in the regression process continued to exhibit some of their ST components, which appeared to be similar to those of the normally forming ST. It has been reported that mammalian male GC undergo apoptosis at 37 °C internal body temperature; birds are the only homeothermic animals to continue spermatogenesis at the increased avian internal body temperature of 40 to 41 °C [45].

ST resorption by apoptosis appeared only to happen as early as 1-wk-old since none was observed afterward, as described in the study. We assume that this could be part of a physiological mechanism participating in the reduction of GC to maintain them in the balance required for normal development of the testes. However, the concrete reasons and factors leading to this resorption are unknown; further investigations of this phenomenon might be required to elucidate this blind spot.

### 4.4. Gonadosomatic Index (GSI) of the Growing Chickens at Different Stages of Growth

Dharani et al. defined the GSI as an indication of the efficiency with which males produce spermatozoa [61]. It is an estimate of gonad weight as a proportion of total body weight, utilized to assess sexual maturity with testicular sexual development [23]. The GSI recorded in this study was about 0.02% among chicks aged between 1-wk-old and 3-wk-old, which was lower than that of the GF: 0.029% at 2-wk-old [23], or that reported among buffalos: 0.04% [62], corresponding to the stage of ST formation and organization on the parenchyma. A decrease in the GSI value was noted at 1-mo-old: 0.01%, which was smaller than that of the GF: 0.039% [23], and showed some similarities to that of the GGC: 0.015%, all in the same age frame [21]. At this time, ST continued their gradual formation displaying a slight increase in testicular weight and body weight, and also primary spermatocytes in a few tubules. A slight increase in this parameter was noted when the chicks reached 1.5-mo-old of age: 0.03%, corresponding to the time when spermatocytes increased in number and became noticeable in many ST. We recorded a GSI of about 0.06% among 2-mo-old chicks, which was double the value of this parameter relative to 1.5-mo-old, significantly smaller than 0.96% for turkeys in the same age frame [24], smaller than that of the GF 0.075% [23], and those reported among certain mammals: 0.078–0.08% in cats, 0.08% among humans, and 0.09% for horses [1,30], but greater than that of the GGC: 0.028% [21]. This increase corresponded to the time when spermatocytes increased in number and were present in more ST. When the chicks reached sexual maturity at 3-mo-old, the GSI was about 0.37% which was the greatest value recorded in this study, corresponding to the time when all spermatogenic cells had developed and become noticeable within the tubular epithelium. This value was greater than those for the GGC and the GF in the same age category: 0.098%, and 0.135% respectively [21,23]. It also exceeded those for certain mammals, e.g., bulls: 0.1% [63], capybara: 0.12%, and collared peccary: 0.21% [1]. It showed great similarities to that of the marmoset: 0.36% but was smaller than those among boars: 0.4%, rats: 0.76%, and mice: 0.55–0.76% [1]. For 4-mo-old specimens, we recorded a GSI of about 0.31%, slightly lower relative to the preceding age point observed but greater than those of the GGC and the GF: 0.106% and 0.138%, respectively [21,23], or compared with the gerbil: 0.22% [64]. The GSI recorded at this age was smaller than those of the NLBC and turkeys in the same age frame: 1.1% and 1.32% respectively [22,24], and those of goats: 0.4% [65], Akodon montensis: 1.1%, and hamsters: 2.13% [1].

The general observations of GSI among chickens in this present study appear consistent with other reported findings, especially those conducted among growing animals that show a gradually increasing trend of GSI, such as turkeys that reached 9.53% at 24-wk-old (6-mo-old) [24]. In this study, GSI showed a greater parameter value in the 3-mo-old chick at sexual maturity, with a slight decrease noted at 4-mo-old. This could mean that greater production of sperm occurs at 3-mo-old, the time at which the testes become fully active and exhibit all types of spermatogenic cells within the tubular epithelium.

## 5. Conclusions

The present study has revealed that the left and right testes’ growth in chickens is almost alike, their increase in length and diameter being related to the spermatogenic activity happening within the ST. Both display SC and spermatogonia as only a single layer of cells from 1-wk-old to 1-mo-old, after which ST complete their formation and allow spermatogonia to differentiate into primary spermatocytes and start displaying more layers of cells. During growth, two great increases in size were recorded, i.e., when the chicks reached 2-mo-old and again at 3-mo-old. The testes exhibited all spermatogenic cells with colonies of spermatozoa noted on the tubular epithelium and also within their lumen, coinciding with sexual maturity. Furthermore, ST resorption by apoptosis occurred in the testes of the 1-wk-old chick and is assumed to be part of the physiological mechanism regulating GC, to maintain GC numbers at a certain level allowing normal growth of the testes. Finally, the highest GSI value was recorded for the 3-mo-old chicks, which could mean that significant sperm production occurs at this time, and that the older the chick grows in age, the higher the GSI becomes.

## Figures and Tables

**Figure 1 vetsci-09-00485-f001:**
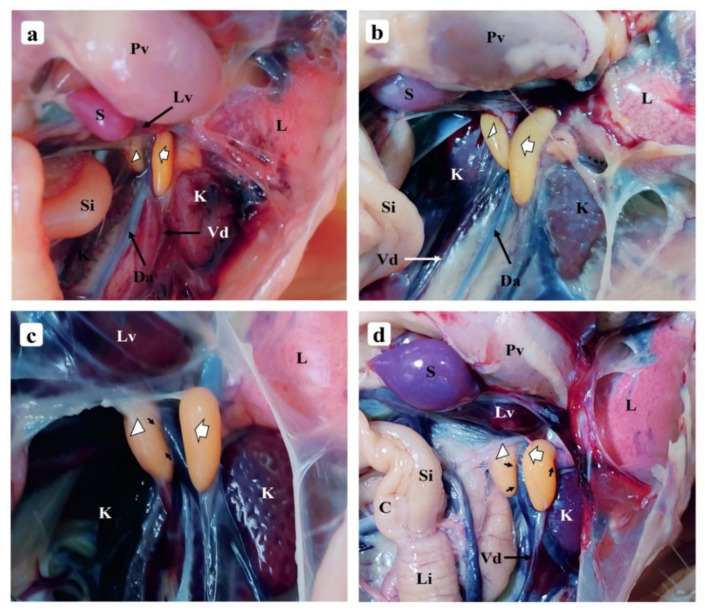
Morphological photographs of chicken testes during growth from 1-wk-old to 1.5-mo-old: (**a**) 1-wk-old; (**b**) 3-wk-old; (**c**) 1-mo-old; (**d**) 1.5-mo-old. Proventriculus (Pv), spleen (S), liver (Lv), the caudal end of the left lung (L), left testis (white arrow), right testis (white arrowhead), small intestine (Si), kidney (K), vas deferens (Vd), dorsal aorta (Da), vascularization growing on the testes surface (black arrows), caecum (C), large intestine (Li).

**Figure 2 vetsci-09-00485-f002:**
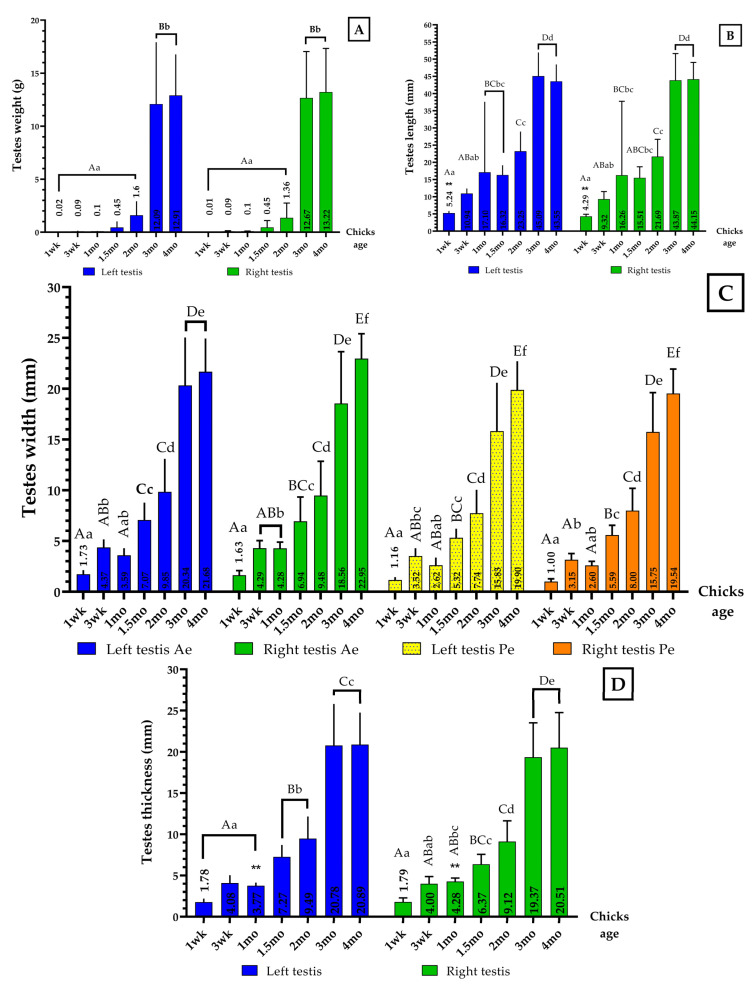
Left and right testes weight, length, width, and thickness evolution at various stages of growth (Mean ± SD). (**A**) Testes weight; (**B**) testes length; (**C**) testes width: Ae: Anterior end of the testis, Pe: Posterior end of the testis; (**D**) testes thickness. Statistical difference: ** *p* < 0.01; Independent samples *t* test for significance. Note: The marked differences are labeled by different superscript lowercase letters (*p* < 0.05); different superscript capital letters indicate extremely significant differences (*p* < 0.01). One-way ANOVA test for significance; Duncan for multiple comparisons.

**Figure 3 vetsci-09-00485-f003:**
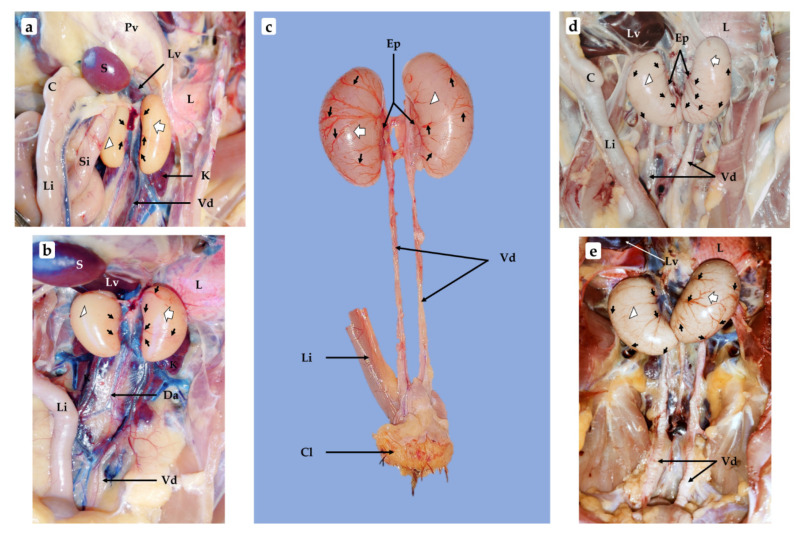
Morphological photographs of chicken testes during growth from 2-mo-old to 4-mo-old: (**a**,**b**) 2-mo-old; (**c**) 3-mo-old; (**d**,**e**) 4-mo-old. Proventriculus (Pv), interior portion of the liver (Lv), spleen (S), caecum (C), large intestine (Li), small intestine (Si), caudal end of the left lung (L), left testis (white arrow), right testis (white arrowhead), vascularization growing on the testes surface (black arrows), left kidney (K), vas deferens (Vd), dorsal aorta (Da), epididymis (Ep), cloaca (Cl).

**Figure 4 vetsci-09-00485-f004:**
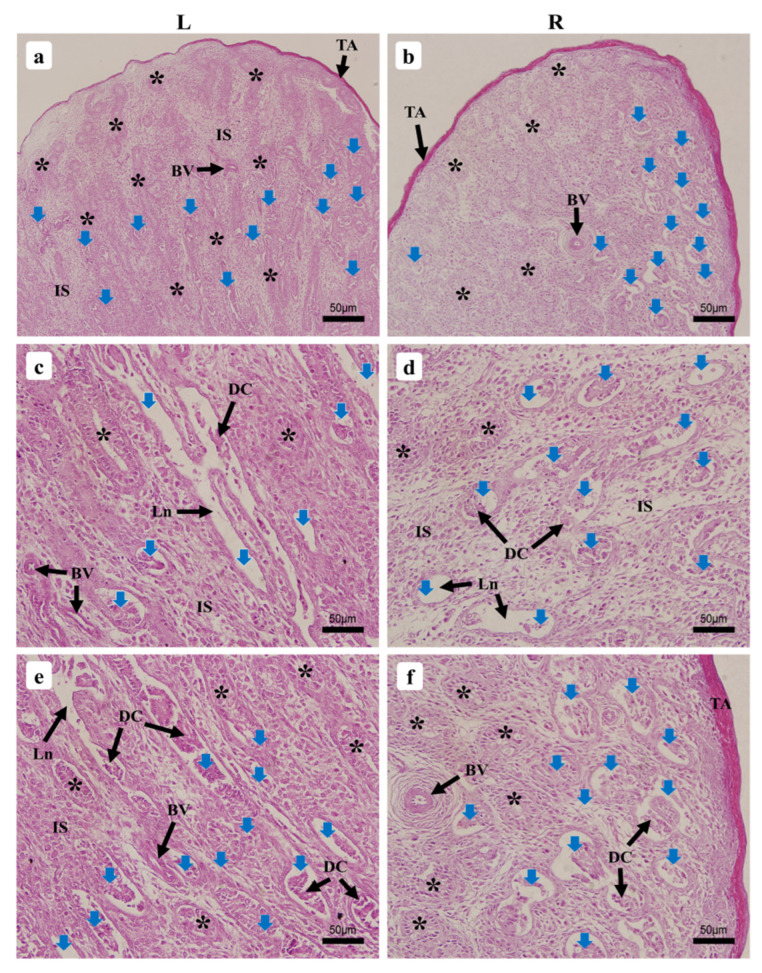
Histological micrographs showing seminiferous tubules’ apoptotic regression within the 1-wk-old chicken testes (**a**,**b**) 40× magnification, (**c**–**f**) 400× magnification. Left testis (L), right testis (R), normally forming seminiferous tubules (*), regressing seminiferous tubules (blue arrows), tunica albuginea (TA), interstitial space (IS), blood vessels (BV), degenerating cells (DC), lacunae (Ln).

**Figure 5 vetsci-09-00485-f005:**
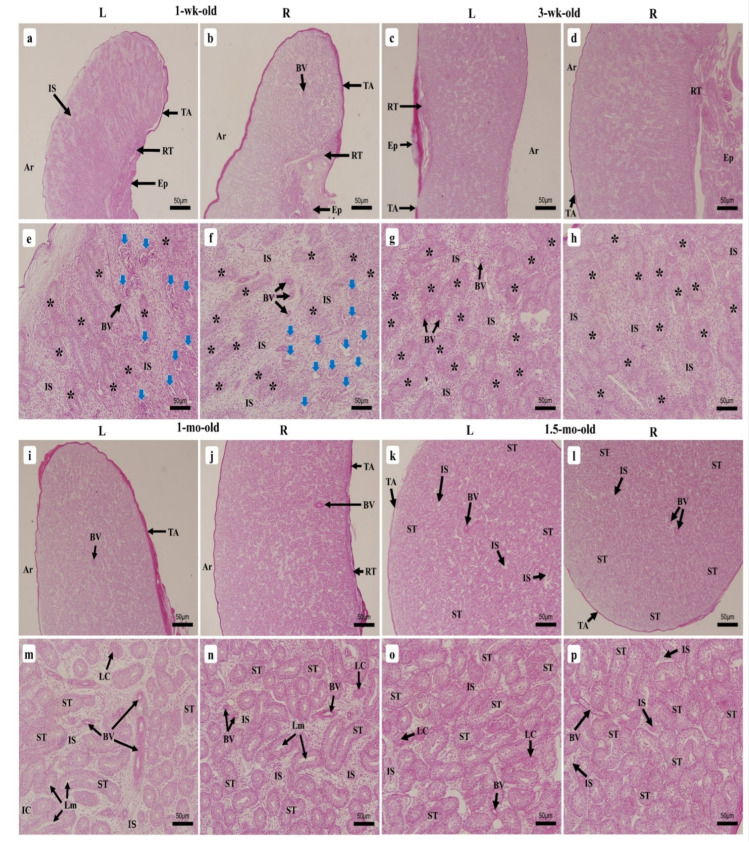
Histological micrographs of chicken testes during growth between 1-wk-old to 1.5-mo-old. (**a**–**d**) and (**i**–**l**) 40× magnification; (**e**–**h**) and (**m**–**p**) 200× magnification. (**a**,**b**,**e**,**f**) 1-wk-old; (**c**,**d**,**g**,**h**) 3-wk-old; (**i**,**j**,**m**,**n**) 1-mo-old; (**k**,**l**,**o**,**p**) 1.5-mo-old. Left testis (L), right testis (R), interstitial space (IS), arch section of the testes (Ar), tunica albuginea (TA), rete testis (RT), epididymis (Ep), blood vessels (BV), emerging seminiferous tubules with unclear delimitation of the basement membrane (*), apoptotic seminiferous tubules on the parenchyma (blue arrows), seminiferous tubules (ST), Leydig cells (LC), lumen (Lm).

**Figure 6 vetsci-09-00485-f006:**
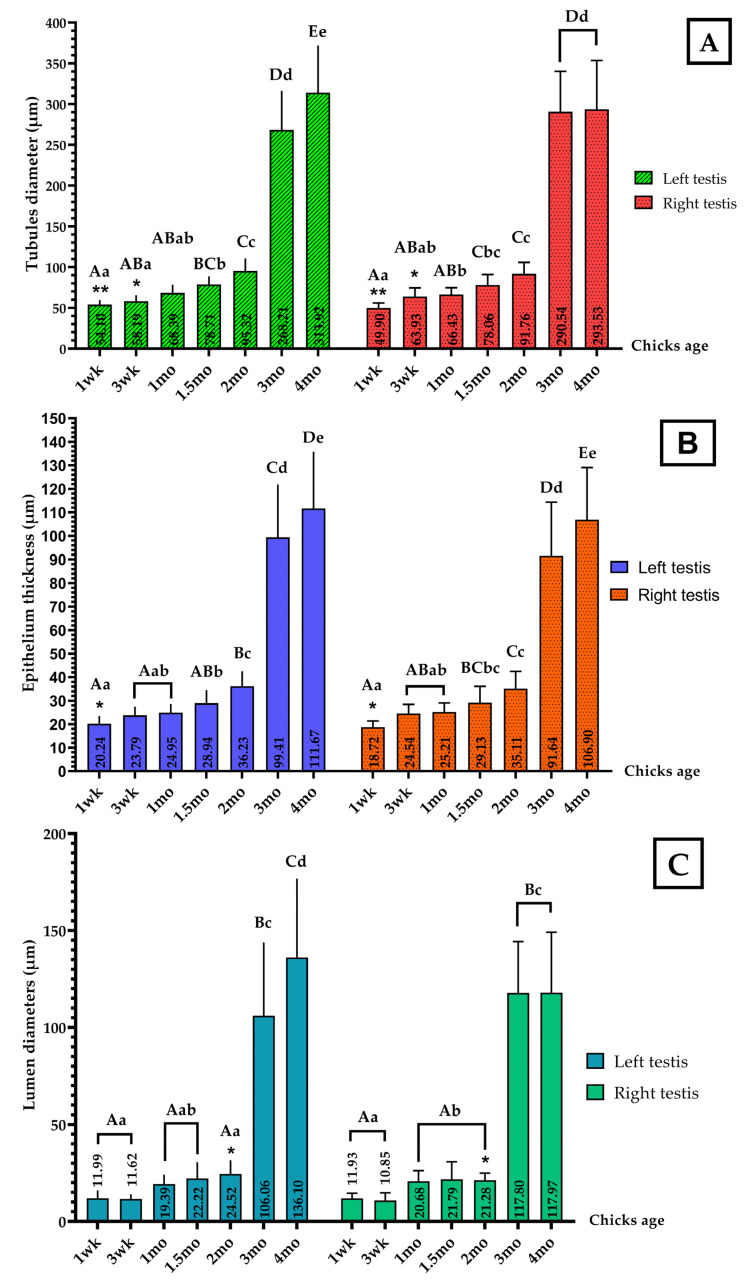
Seminiferous tubules’ morphometry for the growing chickens (Mean ± SD). (**A**) Tubule diameters; (**B**) tubule’s epithelium thickness; (**C**) lumen diameters. Statistical difference: ** *p* < 0.01; * *p* < 0.05; independent samples T tested for significance. Marked differences labeled by different superscript lowercase letters (*p* < 0.05); different capital letters of superscript indicate an extremely significant difference (*p* < 0.01). One-way ANOVA test for significance; Duncan for multiple comparisons.

**Figure 7 vetsci-09-00485-f007:**
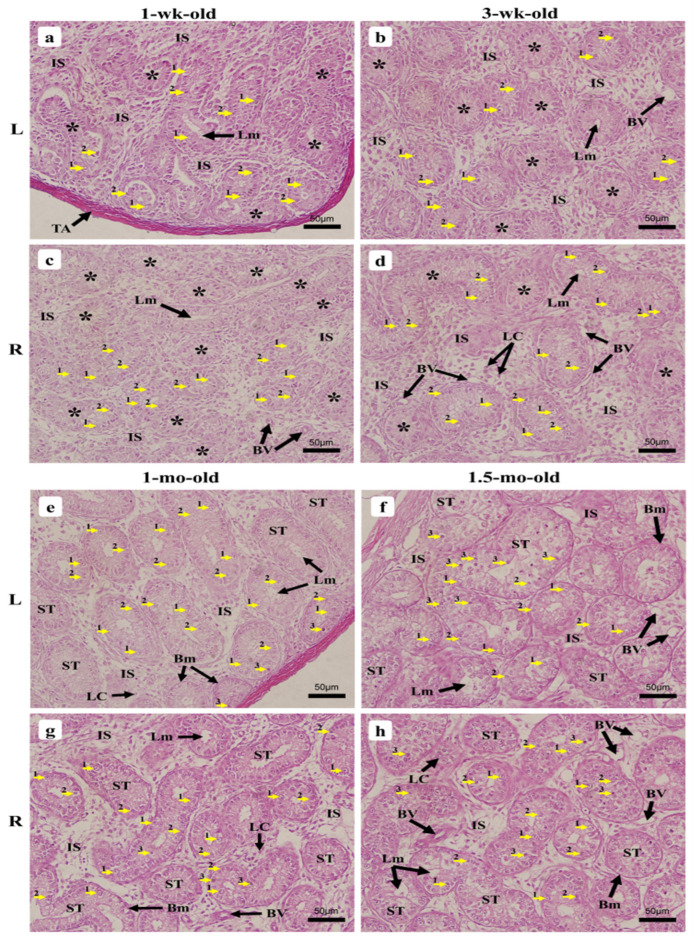
Histological micrographs of chicken testes during growth between 1-wk-old and 1.5-mo-old, 400× magnification. (**a**,**c**) 1-wk-old; (**b**,**d**) 3-wk-old; (**e**,**g**) 1-mo-old; (**f**,**h**) 1.5-mo-old. Left testis (L), right testis (R), tunica albuginea (TA), seminiferous tubules (ST), intertubular space (IS), forming seminiferous tubules with unclear delimitations of their features (*), lumen (Lm), blood vessels (BV), Sertoli cells (1), spermatogonia (2), Leydig cells (LC), spermatocytes (3), basement membrane (Bm).

**Figure 8 vetsci-09-00485-f008:**
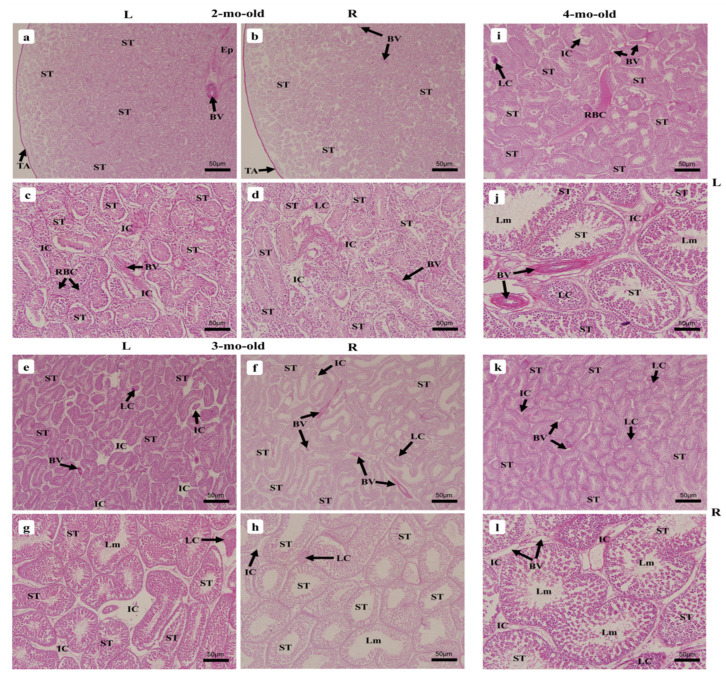
Histological micrographs of chicken testes during growth from 2-mo-old to 4-mo-old. (**a**,**b**,**e**,**f**,**i**,**k**) 40× magnification, (**c**,**d**,**g**,**h**,**j**,**l**) 200× magnification. (**a**–**d**) 2-mo-old; (**e**–**h**) 3-mo-old; (**i**–**l**) 4-mo-old. Left testis (L), right testis (R), tunica albuginea (TA), seminiferous tubules (ST), blood vessels (BV), intertubular compartments (IC), avian red blood cells (RBC), Leydig cells (LC), lumen (Lm).

**Figure 9 vetsci-09-00485-f009:**
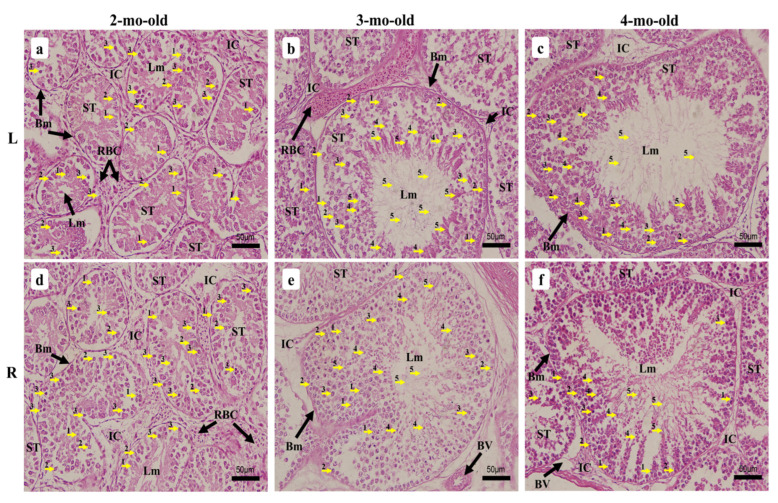
Histological micrographs of chicken testes during growth from 2-mo-old to 4-mo-old, 400× magnification. (**a**,**d**) 2-mo-old; (**b**,**e**) 3-mo-old; (**c**,**f**) 4-mo-old. Left testis (L), right testis (R), basement membrane (Bm), seminiferous tubules (ST), intertubular compartments (IC), avian red blood cells (RBC), lumen (Lm), blood vessels (BV), Sertoli cells (1), spermatogonia (2), spermatocytes (3), spermatids (4), spermatozoa (5).

**Table 1 vetsci-09-00485-t001:** Gonadosomatic index (GSI) of the growing chicks (Mean ± SD).

Parameters	Both Testes Weight (g)	Bodyweight (g)	GSI (%)
wk 1	0.02 ± 0.01 ^Aa^	89.67 ± 7.42 ^Aa^	0.02
wk 3	0.09 ± 0.05 ^Aa^	541.50 ± 66.91 ^Bb^	0.02
mo 1	0.10 ± 0.02 ^Aa^	686.26 ± 70.30 ^Bb^	0.01
mo 1.5	0.45 ± 0.60 ^Aa^	1557.64 ± 134.78 ^Cc^	0.03
mo 2	1.48 ± 1.33 ^Aa^	2361.00 ± 295.88 ^Dd^	0.06
mo 3	12.38 ± 5.02 ^Bb^	3332.00 ± 436.85 ^Ee^	0.37
mo 4	13.07 ± 3.89 ^Bb^	4282.00 ± 419.01 ^Ff^	0.31

Abbreviations: wk: week; mo: month. Statistical difference: Marked difference is labeled by different superscript lowercase letters (*p* < 0.05); different capital letters of superscript indicate an extremely significant difference (*p* < 0.01). One-way ANOVA test for significance; Duncan for multiple comparisons.

## Data Availability

The data presented in this study are openly available in FigShare at: dx.doi.org/10.6084/m9.figshare.20170874 (accessed on 1 July 2022).

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
