# Peer review of "Morpho-Histology and Morphometry of Chicken Testes and Seminiferous Tubules among Yellow-Feathered Broilers of Different Ages"

_vetsci, 2022, doi:10.3390/vetsci9090485_

Round 1

Reviewer 1 Report

This work presents detailed characterization of morphological and histological aspects of chicken testes development. It uses the yellow feather chicken breed and time window 1 week up till 4 month adults.The work is descriptive and extensively long.

Some steps should be taken to shorten the manuscript: e.g. Table 1 and table 2 use identical data, just rearranged for different comparisons - one table (or graphed data should be enough), the same with other tables. Numerical details are repeated in discussion, that should be omitted. Discussion includes parts that are not directly relevant to results and would be appropriate in review article (e.g. lines 435-440).

Minor points:

Abbreviations should be explained at first use: e.g SC (line 45), GSI (line 63) and several other occasions.

AAF solution in Methods should be defined (line 98)

Asterisk (line 295)

Author Response

Here is a point-by-point response to the reviewers’ comments and concerns.

Comment 1: Some steps should be taken to shorten the manuscript: e.g. Table 1 and table 2 use identical data, just rearranged for different comparisons - one table (or graphed data should be enough), the same with other tables.

Response: Agree. Therefore, graphed data have been used instead to present the data of Tables 1, 2, 4, and 5.

Comment 2: Numerical details are repeated in discussion, that should be omitted.

Response: Thanks for this remark. These have been deleted in the reviewed version of the paper.

Comment 3: Discussion includes parts that are not directly relevant to results and would be appropriate in review article (e.g. lines 435-440).

 Response: Thanks for pointing this out. All authors have agreed to remove the related section since it does not affect the other parts of the document which has been done accordingly in the reviewed paper.

Minor points:

Comment 4: Abbreviations should be explained at first use: e.g SC (line 45),

Response: Thank you for pointing this out. This has been defined in the reviewed document.

Comment 5: GSI (line 63) and several other occasions.

 Response: GSI is an acronym that means Gonadosomatic Index, it has already been defined in the abstract (line 15) that is why it has not been written fully in other parts of the manuscript.

Comment 6: AAF solution in Methods should be defined (line 98)

 Response: Thank you for this comment, AAF is an acronym that stands for Alcoholic Acetate Formalin: it’s a solution in which we immersed the collected samples into for fixation before processing them into the paraffine. This has been defined in the abstract and methods sections of the reviewed manuscript where these appear.

Comment 7: Asterisk (line 295)

Response: Thanks for pointing this out, the asterisk was previously written in one word, but it has been changed into its symbol (*) in the reviewed manuscript to keep the consistency with the figures.

Additional clarifications

In addition to the above comments, all spelling and grammatical errors pointed out by the reviewers have been corrected.

Sincerely,

Reviewer 2 Report

Review for “Morpho-Histology and Morphometry of Chicken Testes and Seminiferous Tubules among Yellow-Feathered Broilers of Different Ages”

This report provides detail for a process that is not well described in the current literature.  Therefore, this article with the thorough design and excellent figures is an important addition.  My main concern is that the English usage uneven with some sections hard to follow, particularly in the abstract and introduction.  In contrast, the results and discussions sections are very clear.  I have made a few suggestions below, but I would strongly suggest revision of the abstract and introduction for English usage before publication.

Line 17: define AAF

Line 19: typo “wherefrom” – do you mean “whereas there is a consistent increase in vascularization through 4 months”

First paragraph in Introduction: please specify which animal for which these embryological development dates are listed

Line 69: the “still” in unnecessary.  Suggest rephrasing “…currently available, studies detailing testicular morphology, histology and morphometry of the St and GSI in juvenile chickens is scarce.”

Author Response

Here is a point-by-point response to the reviewers’ comments and concerns.

Comment 1: This report provides detail for a process that is not well described in the current literature.  Therefore, this article with the thorough design and excellent figures is an important addition. 

Response: Thanks a lot for your valuable comments and contribution in the improvement of the quality of this manuscript.

Comment 2: My main concern is that the English usage uneven with some sections hard to follow, particularly in the abstract and introduction. 

In contrast, the results and discussions sections are very clear.  I have made a few suggestions below, but I would strongly suggest revision of the abstract and introduction for English usage before publication.

Response: Thank you very much for your valuable comments that have helped us improve the quality of our manuscript. We have amended the abstract and introduction as pointed in the comments below.

Comment 3: Line 17: define AAF

Response: Thank you for this comment, AAF is an acronym that stands for Alcoholic Acetate Formalin: it’s a solution in which we immersed the collected samples into for fixation before processing them into the paraffine. This has been defined in the abstract and methods sections of the reviewed manuscript where these appear.

Comment 4: Line 19: typo “wherefrom” – do you mean “whereas there is a consistent increase in vascularization through 4 months”

Response: Agree, the meaning is just as you pointed it out and it has been modified in the reviewed manuscript as follows:

‘‘The findings revealed that both testes exhibited a smooth feature from 1-wk-old to 1-mo-old and showed a consistent increase in vascularization thereafter until 4-mo-old.’’

Comment 5: First paragraph in Introduction: please specify which animal for which these embryological development dates are listed

Response: We agree with this and have incorporated the animal (chicken) which embryological development dates are listed in the Introduction of the reviewed version of the manuscript.

Comment 6: Line 69: the “still” in unnecessary.  Suggest rephrasing “…currently available, studies detailing testicular morphology, histology and morphometry of the St and GSI in juvenile chickens is scarce.”

Response: Agree. Therefore, this has been rephrased as follows:

“…studies detailing testicular morphology, histology, and morphometry of the ST, and the GSI in juvenile chickens toward sexual maturity are scarce.”

Additional clarifications

In addition to the above comments, all spelling and grammatical errors pointed out by the reviewers have been corrected.

Sincerely,

Reviewer 3 Report

General Comments:

1. This article records the development of testes of macro and micro levels in broiler chicken

2. Article is well-written but I am concerned about whether the resources and time have been invested in the right direction. There are comparative studies present which have demonstrated a time-lapsed study of testicular development. So, this study adds minimal knowledge to an already existing wide field. 

3. Although I agree that most studies try to interfere with the normal physiological settings, many descriptive anatomical research articles are available. It could have been better to one variable to test the comparative development of target organ.

Specific Comments:

1. Introduction

i. I am unable to understand the benefit this study has over an interval-based study where sampling was performed at regular intervals to observe the spermatogenesis. 

2. Methodology

i. What is AAF?

ii. Can you please give the reference from where you took your sample preparation technique? I am a histologist and there are some inconsistencies in there. Please recheck. 

iii. Can you please give a macros record of your measurements?

2. Results

i. I think it would be much better to give a graphical representation of the results instead of the tabulated results. It gets too congested and the reader feels disconnected altogether until they know exactly what they are looking for. 

3. Discussion

i. Discussion seems to be over-occupied with results and figures which is preventing the formation of a cohesive story. I would suggest rewriting it and focusing more on the bigger picture that it representing rather than re-enlisting the results and citing articles that agree or disagree with your results. 

Author Response

Here is a point-by-point response to the reviewers’ comments and concerns.

General Comments:

Comment 1:

  1. This article records the development of testes of macro and micro levels in broiler chicken
  2. Article is well-written but I am concerned about whether the resources and time have been invested in the right direction. There are comparative studies present which have demonstrated a time-lapsed study of testicular development. So, this study adds minimal knowledge to an already existing wide field.

Response: Thank you very much for this comment.

Comment 2:

  1. Although I agree that most studies try to interfere with the normal physiological settings, many descriptive anatomical research articles are available. It could have been better to one variable to test the comparative development of target organ.

Response: Thank you for such a pertinent comment, we will consider it in our future research projects.

Specific Comments:

Comment 3:

  1. Introduction

I am unable to understand the benefit this study has over an interval-based study where sampling was performed at regular intervals to observe the spermatogenesis.

Response: Thank you for this comment. However, the study did not only aim to observe spermatogenesis, but also evaluate the seminiferous tubules’ morphometry and the Gonadosomatic index among young chicks in addition to paying attention to other details such as the seminiferous tubules apoptotic resorption which seems to have not been reported yet until now.

  1. Methodology

Comment 4: What is AAF?

Response: Thank you for pointing this out. AAF is an acronym that stands for Alcoholic Acetate Formalin: it’s a solution in which we immersed the collected samples into for fixation before processing them into the paraffine. This has been defined in the abstract and methods sections of the reviewed manuscript where these appear.

Comment 5: Can you please give the reference from where you took your sample preparation technique? I am a histologist and there are some inconsistencies in there. Please recheck.

Response: The sample preparation was made according to the following paper:

Okpe, G.C.; Nwatu, U.; Anya, K. Morphometric study of the testes of the Nigerian local breed of chicken. Animal Res. Inter. 2010, 7(2), 1163-1168.

But we used the AAF fixative solution instead since it is what our research team has been working with for years already.

Comment 6: Can you please give a macros record of your measurements?

Response: Here are morphometric raw data.

Testes

W1

W3

M1

M1.5

M2

M3

M4

Weight

(g)

L

0.01

0.15

0.11

0.22

2.55

1.91

15.48

0.02

0.14

0.10

0.42

0.85

1.20

14.75

0.01

0.13

0.08

0.21

1.41

12.79

13.74

0.02

0.16

0.08

0.29

1.09

15.53

17.01

0.01

0.04

0.12

0.37

4.98

14.16

12.35

0.03

0.07

0.11

0.11

0.79

13.64

8.48

0.02

0.07

0.10

0.21

1.75

19.29

7.08

0.02

0.05

0.08

0.30

0.81

14.59

13.69

0.01

0.08

0.12

0.33

0.59

14.21

8.15

0.03

0.03

0.13

2.06

1.15

13.54

18.40

R

0.01

0.14

0.10

0.19

2.03

1.46

14.34

0.01

0.09

0.10

0.41

0.71

11.83

14.03

0.01

0.08

0.09

0.15

0.66

12.11

13.60

0.01

0.13

0.09

0.27

0.70

13.44

19.29

0.01

0.19

0.12

0.36

4.97

14.06

9.81

0.02

0.06

0.12

0.11

0.57

13.78

9.61

0.01

0.07

0.08

0.15

1.98

18.47

7.53

0.01

0.04

0.09

0.25

0.64

14.62

13.85

0.01

0.07

0.11

0.35

0.42

14.62

10.09

0.02

0.04

0.12

2.30

0.89

12.28

20.06

Testes

W1

W3

M1

M1.5

M2

M3

M4

Length

(mm)

L

5.21

11.26

11.62

15.65

29.44

26.98

46.04

5.55

12.85

11.49

16.87

21.79

43.35

49.90

4.81

10.87

10.36

15.64

21.52

46.09

40.94

5.64

11.79

9.56

16.04

26.68

50.23

48.32

4.23

12.75

9.95

19.56

35.64

48.96

43.73

5.66

10.30

10.70

13.69

19.39

44.49

38.74

5.23

11.80

8.74

11.88

21.88

47.75

37.02

5.38

9.12

75.40

15.98

18.24

44.38

45.34

4.40

9.02

10.36

15.71

17.78

51.19

36.73

6.31

9.59

12.79

22.13

20.13

47.51

48.73

R

4.42

10.06

9.88

13.40

23.81

23.71

47.75

3.98

10.61

9.79

16.84

21.21

41.84

45.03

3.75

7.67

9.85

12.75

21.91

46.17

42.98

4.63

9.71

9.31

15.14

20.66

46.70

48.61

4.45

13.91

9.93

18.81

32.26

50.66

39.31

5.46

6.70

8.49

13.30

20.64

44.85

40.61

3.92

8.92

8.73

10.85

26.90

51.75

35.73

3.61

6.27

77.40

16.90

15.67

45.30

44.02

3.67

9.96

9.10

15.39

17.14

45.67

44.68

5.00

9.35

10.11

21.79

16.74

42.08

52.75

Testes

W1

W3

M1

M1.5

M2

M3

M4

Width Anterior End (mm)

L

1.20

5.24

3.56

6.14

14.92

10.80

25.68

1.66

3.84

2.16

6.77

9.20

20.10

21.34

1.70

3.87

2.66

6.24

10.73

22.41

20.55

2.14

4.71

3.44

5.43

6.62

15.08

20.85

1.17

6.08

3.64

7.69

15.04

24.11

22.34

2.00

3.87

4.46

5.48

9.84

24.93

18.47

2.24

4.19

4.13

7.68

10.96

26.48

15.62

1.70

3.31

3.90

6.59

5.44

20.43

23.31

1.31

4.51

3.94

7.30

6.75

20.29

21.74

2.16

4.09

4.04

11.37

9.04

18.77

26.88

R

1.13

4.57

4.38

5.72

14.53

10.36

20.97

1.48

4.00

3.58

7.44

8.81

22.73

18.79

1.28

4.58

3.34

5.45

9.02

18.67

24.07

1.68

5.31

4.41

6.61

5.90

24.44

23.95

1.37

5.27

5.07

6.58

16.62

23.06

21.81

2.57

3.63

4.97

4.64

8.17

17.68

24.44

1.70

4.96

3.77

6.32

8.48

24.06

21.12

2.25

3.27

4.11

7.31

7.78

13.25

25.90

1.49

3.52

4.13

5.98

7.05

18.64

21.76

1.36

3.79

5.00

13.34

8.42

12.75

26.72

Testes

W1

W3

M1

M1.5

M2

M3

M4

Width Posterior End (mm)

L

1.01

3.68

2.53

4.85

9.61

6.67

22.38

1.08

3.36

1.71

5.49

7.43

18.53

20.31

1.02

3.15

2.22

4.97

9.41

16.34

19.67

1.00

3.85

3.11

6.05

6.25

19.99

23.52

0.71

5.34

2.91

4.41

12.81

12.61

18.59

1.75

2.55

2.54

3.74

7.01

14.27

17.33

1.44

3.28

3.86

5.97

7.82

24.56

14.91

1.19

2.80

3.18

5.56

5.35

15.74

22.93

1.12

4.00

2.86

6.92

6.24

16.15

17.58

1.23

3.20

1.28

5.28

5.42

13.46

21.80

R

0.79

3.49

2.38

5.26

9.99

7.01

20.77

1.17

1.86

2.74

5.95

5.89

17.17

18.33

0.77

3.97

1.97

4.86

9.23

15.64

20.93

1.18

3.30

2.95

5.28

8.23

18.84

20.74

1.20

3.61

2.33

5.72

12.70

16.51

18.79

1.56

3.14

3.12

4.30

6.72

15.23

18.29

0.71

2.86

2.76

4.60

8.46

21.82

16.81

1.02

2.46

2.35

5.84

6.35

17.00

19.89

0.71

3.12

2.27

7.47

6.25

15.30

16.33

0.98

3.66

3.15

6.62

6.13

12.98

24.60

Testes

W1

W3

M1

M1.5

M2

M3

M4

Thickness (mm)

L

1.58

3.79

3.93

5.78

11.40

10.91

28.93

1.78

3.33

3.37

6.78

9.04

23.62

19.89

1.38

4.15

3.37

6.42

11.26

22.14

22.07

2.62

4.22

3.29

7.02

8.34

25.96

23.64

1.58

6.46

3.65

8.56

15.39

23.75

19.87

2.29

4.46

3.99

5.61

8.51

13.79

17.71

1.99

4.21

3.90

7.79

10.01

26.57

15.01

1.78

3.01

3.95

7.86

6.93

20.92

21.10

1.27

3.52

4.49

6.56

6.23

20.22

17.73

1.51

3.62

3.74

10.32

7.79

19.88

22.98

R

1.19

5.10

4.29

5.20

12.12

9.90

26.68

1.62

3.28

4.16

7.00

8.16

20.48

15.72

1.37

4.80

3.74

6.04

9.39

17.32

18.54

2.13

4.08

4.11

6.84

6.50

21.23

26.77

1.65

5.10

3.82

6.31

14.59

21.67

17.03

2.49

4.36

4.44

4.89

9.18

17.34

19.79

1.92

4.01

3.98

5.44

8.91

26.13

16.11

2.53

2.47

4.37

6.64

7.64

19.74

21.92

1.88

3.44

5.02

6.18

6.37

19.91

17.83

1.15

3.35

4.89

9.15

8.33

20.00

24.66

  1. Results

Comment 7: I think it would be much better to give a graphical representation of the results instead of the tabulated results. It gets too congested and the reader feels disconnected altogether until they know exactly what they are looking for.

Response: Agree. Therefore, graphed data have been used instead to reduce the Tables number.

  1. Discussion

Comment 8: Discussion seems to be over-occupied with results and figures which is preventing the formation of a cohesive story. I would suggest rewriting it and focusing more on the bigger picture that it representing rather than re-enlisting the results and citing articles that agree or disagree with your results.

Response: Thank you for this suggestion. The discussion has been reviewed in some parts of its content.

Additional clarifications

In addition to the above comments, all spelling and grammatical errors pointed out by the reviewers have been corrected.

Sincerely,

Round 2

Reviewer 1 Report

comments from review were addressed by the authors

Author Response

Review report to Academic Editor

Dear Dr. Patrick Butaye,

Thank you for giving us the opportunity to submit a revised draft of our manuscript titled Morpho-Histology and Morphometry of Chicken Testes and Seminiferous Tubules among Yellow-Feathered Broilers of Different Ages to Veterinary Sciences. We appreciate the time and effort that you and the reviewers have dedicated to providing your valuable feedback on our manuscript. We are grateful to the reviewers and the academic editor for their insightful comments on our paper. We have been able to incorporate changes to reflect most of the suggestions provided by the Academic Editor. We have activated the Track changes feature within the manuscript.

Here is a point-by-point response to the Academic Editor's comments and concerns.

Comment 1: There are a great variety of chicken breeds, or, in commercially kept chickens, of chicken lines which, depending on their kind of use (broilers, laying hens, parent or grandparent lines, backyard chickens) differ in a great extent regarding the time they need to reach sexual maturity. I would therefore like to ask you to add more information on the chickens called Yellow-Feathered Broilers which might not be generally known to the international readership of Veterinary Sciences.

Response: Agree and thank you for this suggestion. Therefore, more information about the Chinese yellow-feathered broilers has been added in the revised version of the manuscript: Materials and methods (Line 101-104) and in the Discussion (Line 469-488).

Comment 2: Is this a commercial chicken line used for meat production? If yes, is it a fast-growing line (at which age do they reach their slaughtering weight)?

Response: They are indeed native commercial chickens well-known in Asia, particularly in China where they account for 50% of the total chicken meat production.

These chickens are generally classified into three groups according to their growth rate: (1) Fast growth type: ~60 days, (2) Medium speed type: 60-100 days, (3) and slow growth type: more than 100 days with a slaughtering weight generally about 1.75 Kg.

The chickens used in this study belong to the slow growth type and are generally slaughtered at the age of 3-month-old.

However, little information on this strain can be found online, especially in English. The few available are mostly in Chinese: https://www.jbzyw.com/view/324204

Comment 3: Or alternatively, is this a parent line?

 Response: Yellow-feathered chickens are Chinese native broilers, which are bred by crossbreeding of local high-quality breeds.

Comment 4: To my opinion there is a need for more information to the question whether the results regarding the developmental speed of the testes can be transferred to other broiler lines or to males of laying hen lines. Or to backyard chickens.  

Response: Thank you for this suggestion. Therefore, we cannot confirm that because to date, no manuscripts have been found reporting related results. This is a valuable suggestion that could be used as a whole research project.

Comment 5: It would therefore be good to have some information in the Material and Method section and in the discussion about this topic.

 Response: Thank you for this suggestion. More information on this strain of chickens has been included in the reviewed version of the manuscript.

Additional clarifications

In addition to the above comments pointed out by the Academic Editor, a Simple Summary has been added to the reviewed version of the paper. We also did an update on certain acronyms that now appear in the Simple Summary as well as the reference list.

We look forward to hearing from you in due time regarding our submission and to responding to any further questions and comments you may have.

Sincerely,

Reviewer 3 Report

Dear Authors, 

You have made significant improvements in your draft. However, I am sorry to say that I am still not very convinced about whether the study design is appropriate. In this day and age, I would recommend adding at least one more specie to compare them side by side and resubmit your article after that. 

I am leaving this decision with the Editor. I would reiterate here that you have made significant improvements but I am not convinced of its scientific significance in its current form. 

Best of luck with your manuscript. 

Author Response

(The authors gave the same response as above.)
